# Vitamin D in Triple-Negative and BRCA1-Deficient Breast Cancer—Implications for Pathogenesis and Therapy

**DOI:** 10.3390/ijms21103670

**Published:** 2020-05-23

**Authors:** Janusz Blasiak, Elzbieta Pawlowska, Jan Chojnacki, Joanna Szczepanska, Michal Fila, Cezary Chojnacki

**Affiliations:** 1Department of Molecular Genetics, Faculty of Biology and Environmental Protection, University of Lodz, 90-236 Lodz, Poland; 2Department of Orthodontics, Medical University of Lodz, 92-216 Lodz, Poland; elzbieta.pawlowska@umed.lodz.pl; 3Department of Clinical Nutrition and Gastroenterological Diagnostics, Medical University of Lodz, 90-647 Lodz, Poland; jan.chojnacki@umed.lodz.pl (J.C.); cezary.chojnacki@umed.lodz.pl (C.C.); 4Department of Pediatric Dentistry, Medical University of Lodz, 92-216 Lodz, Poland; joanna.szczepanska@umed.lodz.pl; 5Department of Neurology, Polish Mother Memorial Hospital Research Institute, 93-338 Lodz, Poland; michalfila@poczta.onet.pl

**Keywords:** vitamin D, triple-negative breast cancer, BRCA1, genomic stability, DNA repair

## Abstract

Several studies show that triple-negative breast cancer (TNBC) patients have the lowest vitamin D concentration among all breast cancer types, suggesting that this vitamin may induce a protective effect against TNBC. This effect of the active metabolite of vitamin D, 1α,25-dihydroxyvitamin D3 (1,25(OH)2D), can be attributed to its potential to modulate proliferation, differentiation, apoptosis, inflammation, angiogenesis, invasion and metastasis and is supported by many in vitro and animal studies, but its exact mechanism is poorly known. In a fraction of TNBCs that harbor mutations that cause the loss of function of the DNA repair-associated breast cancer type 1 susceptibility (*BRCA1*) gene, 1,25(OH)2D may induce protective effects by activating its receptor and inactivating cathepsin L-mediated degradation of tumor protein P53 binding protein 1 (TP53BP1), preventing deficiency in DNA double-strand break repair and contributing to genome stability. Similar effects can be induced by the interaction of 1,25(OH)2D with proteins of the growth arrest and DNA damage-inducible 45 (GADD45) family. Further studies on TNBC cell lines with exact molecular characteristics and clinical trials with well-defined cases are needed to determine the mechanism of action of vitamin D in TNBC to assess its preventive and therapeutic potential.

## 1. Introduction

Studies on the anticancer properties of vitamin D bring conflicting results and the reason for this conflict has not yet been identified (reviewed in [1,2]). Many factors influence vitamin D status in the human body, including sunlight exposure, diet, supplements and lifestyle. On the other hand, cancer represents a diverse group of diseases that often have little in common. Therefore, a thesis on the general anticancer properties of vitamin D is not fully justified, especially since the molecular mechanism beyond such properties is not completely known. Consequently, further studies on the anticancer properties of vitamin D should include the precise determination of vitamin D status, both in in vivo and in vitro studies, and focus on a well-defined cancer type and a specific stage of cancer transformation.

Many studies that have been performed so far to evaluate the anticancer potential of vitamin D include three kinds of malignancies: breast, prostate and colon cancers (reviewed in [2]). Although there are some common features among them, e.g., the first two represent hormone-dependent cancers, each of them signifies a different, not fully known, pathway of pathogenesis. Consequently, they may represent different reactions to vitamin D.

Breast cancer is the most common malignancy in women and one of the three most common cancers worldwide with therapy substantially progressing in the last years. Radical primary surgery is not an optimal treatment option for a significant proportion of breast cancer patients [3]. Triple negative breast cancer (TNBC) is characterized by lack of expression of estrogen receptors (ER) and progesterone receptors (PR) and lack of overexpression or amplification of human epidermal growth factor 2 (HER2). TNBC certainly has a worse therapeutic outcome than other breast cancer types [4]. Therefore, new therapeutic options and improvements of the existing ones are needed. Although defined in many aspects, TNBC is not homogenous, but represents several diverse molecular and clinical features and at least six distinct subtypes of TNBC can be considered [5]. Moreover, TNBC often displays molecular and clinical characteristics similar to DNA repair-associated breast cancer type 1 (BRCA1)-deficient breast cancer cases [6].

Several excellent reviews on the involvement of vitamin D in cancer transformation have been published (e.g., [1,2,7,8,9] and references therein). This manuscript summarizes and updates recent evidence on the effects of vitamin D in breast cancer with a special emphasis on its triple-negative variants that are frequently similar to BRCA1-deficient cases.

## 2. Vitamin D—an Essential Nutrient and a Potential Preventive Agent and Pharmaceutical in Breast Cancer

Humans can endogenously synthesize vitamin D_3_ (vitamin D) in the skin when exposed to UVB present in solar radiation. However, their current lifestyle and environmental conditions make that exposure insufficient for the production of the required amount of vitamin D, which, in turn, implies the need for its dietary supplementation and makes it an essential human micronutrient (reviewed in [10]).

1α,25-dihydroxyvitamin D3 (1,25(OH)2D, calcitriol) is a biologically active metabolite of vitamin D and an endocrine hormone that binds to its receptors: transmembrane membrane-associated rapid response steroid-binding (MARRS) and cytosolic vitamin D receptor (VDR), which is a member of the nuclear receptor superfamily of ligand-inducible transcription factors [11,12] (Figure 1). This 1,25(OH)2D-VDR complex is often associated with retinoid X receptor alpha (RXRA), assisted by steroid receptor coactivator 1 (SRC-1) and together they bind vitamin D response elements (VDREs) in the promoters of genes they regulate to activate or repress their transcription [13]. Downstream targets of VDR are genes of mineral metabolism, including calcium and phosphate homeostasis and many other metabolic pathways, including those involved in the immune response and cancer (reviewed in [14]). It was shown that heterodimerization and the recruitment of the VDR/RXRA complex to target genes in hepatic cells is promoted by sequestosome 1 (p62/SQSTM), a key autophagy receptor [15].

The serum concentration of the stable precursor of 1,25(OH)2D, 25-hydroxyvitamin D (25(OH)D, calcidiol), is usually used as a marker of the actual concentration of vitamin D in organisms.

Synthesis of vitamin D in the skin requires UV radiation, which can induce DNA damage, preferentially cyclobutane pyrimidine dimers and (6-4)-pyrimidine-pyrimidone photoproducts, which are found in many skin cancers, including melanoma (reviewed in [16]). Vitamin D protects the skin against UV-induced aging and damage in many pathways that may save it from UV-induced DNA damage and resulting skin cancer (reviewed in [16]). This raises a question about the general anti-cancer properties of vitamin D.

Usually, the biological effects of vitamin D are categorized into non-genomic and genomic (reviewed in [17]). The genomic effects of 1,25(OH)2D, as depicted in Figure 1, are mediated by VDR, RXR and VDREs and result in long-term, sometimes delayed, biological consequences. Some rapid, sometimes transient, cellular effects of 1,25(OH)2D through non-genomic pathways are known. One of them is the protection of VDR-defective human fibroblasts against DNA damage induced by UV mediated by the endoplasmic reticulum stress protein 57 (ERP57, MARRS) [18]. Therefore, the anticancer effects of vitamin D may be underlined by both genomic and non-genomic actions as UV-induced DNA damages, including cyclobutane dimers and (6-4)-photoproducts, are observed in skin cancers (reviewed in [19]).

Colston et al., and Abe et al., were the first to show the directly anticancer properties of vitamin D in vitro [20,21]. Then, many in vivo and in vitro studies showed the protective action of vitamin D against various cancers and several genes important in cancer transformation were identified as targets in the genomic action of this vitamin (reviewed in [2]). In general, the antineoplastic effects of vitamin D are mainly underlined by its involvement in the regulation of specific signaling pathways that direct cancer growth. In estrogen receptor-positive (ER+) cases of breast cancer in postmenopausal women, tumor growth is maintained by the local production of estrogen due to the lack of ovarian synthesis of this hormone [22]. Vitamin D and its analogues were reported to selectively inhibit aromatase, a key enzyme in estrogen synthesis, and downregulate estrogen receptor alpha (ERα) in breast tissue, so they can be considered in the prevention and therapy of postmenopausal ER-positive breast cancer cases [23,24,25,26,27,28].

The anticancer effects of 1,25(OH)2D can be generally attributed to is potential to modulate proliferation, differentiation, apoptosis, inflammation, angiogenesis, invasion and metastasis (reviewed in [1]). The essential role of VDR in mediating vitamin D anticancer effects in TNBC was shown by LaPorta and Welsh, who demonstrated that 1,25(OH)2D downregulated genes related to breast cancer invasion and metastasis in cells from a mouse model of TNBC [29]. However, these effects were not observed in mice with VDR knockout, but the reintroduction of human VDR restored the ability of 1,25(OH)2D to inhibit the proliferation of TNBC-like cells. Therefore, VDR is necessary to mediate effects of vitamin D in TNBC cells.

## 3. Vitamin D in Breast Cancer

1,25(OH)2D is involved in the regulation of the proliferation and differentiation of normal mammary gland cells through VDR signaling [30]. Therefore, disturbance in this signaling may result in an aberrant proliferation typical of cancer cells. This is supported by the research on VDR knockout mice, which displayed aberrant ductal differentiation and branching in the mammary gland [31,32]. These and other studies suggest antiproliferative and differentiating properties of 1,25(OH)2D and its use as an anticancer agent [21,33]. However, both primary cancer cell cultures and cancer cell lines are often characterized by an insensitivity to vitamin D. On the other hand, Friedrich et al., observed the strong immunochemical reactivity of VDR in breast cancer specimens from 228 patients, although no correlation was found between VDR expression and tumor stage, lymph node status, grading, tumor type, expression of estrogen receptors or progesterone receptors (PRs), the proliferation markers Ki-67 and p53 or the S-phase index [34]. These authors concluded that VDR cannot be considered as a strong prognostic factor in breast cancer. However, El-Azhiri et al., showed that majority of 1114 breast cancer samples displayed the strong or moderate immunochemical reactivity of VDR, whose expression was negatively correlated with tumor size, hormonal receptor, triple-negative status and Ki-67 [35]. Still, VDR expression was not associated with survival time. Altogether, these results suggest that vitamin D3 treatment can be effective against aggressive breast cancers. It is not easy to explain the different outcomes of the results of Friedrich et al., and El-Azhiri et al., but tumor heterogeneity might play a role. However, El-Azhiri et al., did not consider their results as a proof of evidence.

As mutations in the *VDR* gene are rare in cancer, Marik et al., focused on the epigenetic regulation of this gene in breast cancer [36]. They detected hypermethylated CpG islands upstream and near the transcription start site of the *VDR* gene and their demethylation resulted in an increase in VDR mRNA levels in breast cancer cell lines. Primary breast tumor cells displayed hypermethylation in these islands in contrast with normal mammary cells. However, VDR mRNAs in breast cancers were 5′-trunctated in most cases. Parallel to this, genes containing VDREs were underexpressed in breast cancer. The treatment of breast cancer cells with 5-aza-2′-deoxycytidine (5-aza-dC), a DNA demethylation agent, restored the full length *VDR* transcript in breast cancer cells. Therefore, chemical demethylation of the *VDR* gene may reverse 1,25(OH)2D insensitivity in breast cancer cells and make them prone to various 1,25(OH)2D-based therapies.

It is accepted that VDR in breast cancer cells is necessary and sufficient for the tumor-suppressive effects of vitamin D (reviewed in [9])**.** In general, VDR activation in breast cancer may result in the inhibition of the cell cycle, cell death and the induction of differentiation in breast cancer cells [1,33]. Cell death may be executed by apoptosis or autophagy, but recent studies demonstrated that autophagy may occur in luminal, but not in basal breast cancer cells [37]. It was also shown that vitamin D supplementation increased autophagy in mouse mammary glands.

Many observational and clinical studies showed the ability of vitamin D to modulate cancer induction and progression (reviewed in [1,7,9]). These studies focused on the role of VDR and its downstream gene products, including cytochrome P450 family 27 subfamily B member 1 and family 24 subfamily A member 1 (CYP27B1 and CYP24A1) in breast cancer progression. Vitamin D status was assessed by the concentration of 25(OH)D and 1,25(OH)2D, UV exposure, dietary intake and supplementation, variability of the genes of the vitamin D pathway and others.

O’Brien et al., found that DNA methylation of CpGs in vitamin D-related genes with a potential link to immune function genes might interact with 25(OH)D in serum to modify the risk of breast cancer in large case–control cohort studies [8].

In a meta-analysis comprising nine prospective studies with 5206 cases and 6450 controls, Bauer et al., showed a nonlinear inverse association between breast cancer risk in postmenopausal women and plasma vitamin D levels measured as the 25(OH)D concentration [38]. This relationship was not modulated by invasiveness, body mass index, anatomical region, postmenopausal hormone use, or the assay method. Therefore, the relationship between breast cancer risk and plasma vitamin D concentration may be strongly affected by menopause. This may explain the inconsistency in the results obtained and have significant clinical implications. This study confirms earlier research by Crew et al., who showed a similar correlation in 1026 cases and 1075 controls [39]. They showed that plasma 25(OH)D was inversely associated with breast cancer risk in a concentration-dependent manner. The cases of TNBC occurring in pre-menopausal women are usually associated with a rich mutational landscape and the use of hormonal contraceptives (reviewed in [40]).

Around a 30% decrease in breast cancer risk was observed in women with the highest quantiles of circulating 25(OH)D compared with women with the lowest quantiles in the Nurses’ Health Study (NHS), a part of the Harvard cohort studies [41].

The results of epidemiological studies on vitamin D and breast cancer do not allow us to draw a definite conclusion on the possible correlation between breast cancer occurrence and vitamin D intake or a lack thereof. It likely depends on several factors, primarily menopausal status. Moreover, a case–control study should be standardized for sunlight exposure as “cases” are expected to be less exposed due to the diseases forcing them to reduce their physical and outdoor activity.

In summary, the effects of vitamin D in breast cancer depend on its receptor, but VDR signaling is highly heterogeneous and incompletely known both in normal mammary glands and breast tumors. This seems to be especially important in that some cells are vitamin D sensitive and some are resistant within a single tumor. Moreover, vitamin D resistance may occur during cancer progression due to epigenetic changes in the *VDR* gene or expression of other genes including *CYP24A1* [42]. Therefore, the results of many studies should be re-interpreted due to this VDR heterogeneity. Moreover, vitamin D status in some specific phases in breast development, including pregnancy and menopause, should be taken into account. However, despite these objections, further studies on the role of vitamin D in breast cancer prevention and therapy need to be continued as it is estimated that about a half of all cases can be vitamin D-responsive [9].

## 4. Vitamin D3 Signaling in Triple-Negative Breast Cancer

### 4.1. Observational Studies

Lope et al., analyzed the association between serum 25(OH)D and breast cancer risk in 546 histologically confirmed breast cancer cases of various subtypes and 558 controls [43]. They observed a protective effect of 25(OH)D against breast cancer risk, with a significant dose–response trend, and the protection was more pronounced for TNBC tumors than other cases. Similar results were obtained by Peppone et al., who concluded that breast cancer patients with a more aggressive cancer phenotype, including TNBC tumors and worse prognoses, had lower mean 25(OH)D concentrations than those with more favorable prognoses and less aggressive phenotypes, as well as non-cancer controls [44].

Kim et al., observed that serum concentration of 25(OH)D was inversely associated with the prognosis of patients with breast cancer of the luminal A and luminal B subtypes, but not with HER2-enriched or TNBC subtypes [45]. In a similar study, it was observed that the TNBC phenotype was characterized by the lowest serum 25(OH)D level and the highest fraction of patients that were vitamin D deficient [46]. It was concluded that the TNBC phenotype was associated with low vitamin D levels.

In a systematic review, Tommi et al., tried to establish a relationship between serum levels of vitamin D (25(OH)D) and the occurrence of TNBC [47]. They analyzed 14 studies with the cumulative results of 13,135 breast cancer cases. When these cases were grouped according to breast cancer type, the fraction showing a significant association between vitamin D status and breast cancer occurrence was 37% for non-TNBC, 48% for analyses that included some TNBC cases, and 88% for TNBC. These results suggest that low serum vitamin D level may be associated with an increased risk of TNBC occurrence.

Viala et al., analyzed the level of vitamin D in breast cancer patients after surgery and during or after neoadjuvant chemotherapy [48]. They noted that deficiency in vitamin D (less than 20 ng/mL 25(OH)D in serum) was not associated with survival in the entire studied population of 327 women, but a positive trend was observed in TNBC cases and in the hormone receptor-positive/HER2-negative subgroup. Higher serum levels of 25(OH)D were associated with a reduced risk of breast cancer, especially in cases with poor prognosis, including TNBC [49].

As sunlight exposure significantly contributes to the synthesis of vitamin D in the skin and the level of this vitamin is associated with the aggressiveness of breast cancer, it seems reasonable to look for a possible correlation between the intensity of sunlight exposure and TNBC occurrence. It was shown that the ratio of TNBC tumors among breast cancer patients from a high sunlight exposure area in Turkey was 12%, whereas such a ratio was only 8% in patients from a region with considerably lower sunlight exposure [50]. Although the results of this study are not in line with the general conception of the protective effect of vitamin D against TNBC occurrence, it presents only an association without an analysis of several factors and parameters that may be related to vitamin D metabolism and also vitamin D concentration.

It was shown that increased sun exposure was associated with a reduced risk of TNBC in African American/black women [51]. It is known that breast cancer occurrence in the United States is highest in women of European ancestry, but African American women have the highest ratio of the cancer diagnosed in young age and the highest occurrence of the most aggressive types of breast cancer, including TNBC [52,53,54]. It is not completely clear what stands behind this difference. Africans, especially those living in sub-Saharan regions, have sufficient levels of vitamin D, but, when living in North America, often present a deficiency in this vitamin [55,56]. Dark skin with high melanin content is hypothesized as the ancestral skin color of origin and the adaptation to lesser sunlight exposure results in lighter skin and consequently lesser synthesis of vitamin D. Therefore, it is justified to hypothesize that low vitamin D levels might be associated with a higher prevalence of more aggressive tumors among African American women than women of European ancestry. The problem of the preventive potential of vitamin D in TNBC from the perspective of evolution and adaptation in association with racial disparities was reviewed by Yao et al. [57]. In general, studies on the association between breast cancer, vitamin D intake and sun exposure are inconsistent and this inconsistency likely results from considering breast cancer as a homogenous disease. Recent studies on genome-wide gene expression profiling strengthened our view on the complexity and diversity of TNBC, suggesting at least six different molecular subtypes of TNBC, namely basal-like 1 (BL1), basal-like 2 (BL2), immunomodulatory (IM), mesenchymal (M), mesenchymal stem-like (MSL) and luminal androgen receptor (LAR) [58]. As we cited above, many studies point at the most aggressive breast cancer types, including TNBC, as those most strongly associated with low vitamin D levels, whereas other types may give a weak association or no association at all.

Insulin-like growth factor receptor 1 (IGFR) is considered as a potential target in cancer therapy (reviewed in [59]). Soljic et al., correlated VDR and IGFR expression with clinico-pathological characteristics of TNBC patients [60]. They observed a positive VDR immunostaining in 27% of cases that negatively correlated with mitotic score, histological grade and a higher proliferation index, evaluated by the expression of Ki-67 and increased overall survival (OS). Over 35% of patients displayed IGFR expression and this positively correlated with a higher mitotic score, Ki-67, and shorter disease-free survival (DFS). Patients who had VDR-negative and IGF-positive tumors were characterized by lower DFS and OS. This study confirms that about one third of TNBC tumors express VDR and/or IGFR and this expression can be linked with both molecular and clinical parameters that can be useful in stratifying patients for therapy.

### 4.2. Oxidative Stress, Proliferation, Differentiation and Inflammation

It was shown that menadione (MEN), a glutathione depleting agent, potentiated growth arrest induced by 1,25(OH)2D in breast cancer cell lines through ROS generation, leading to DNA damage [61,62]. This effect was confirmed in vivo in the syngeneic host of murine-transplantable triple-negative breast tumor M-406 [63].

The aggressiveness of TNBC may be associated with inflammatory processes characterized by deregulation of the immunological responses with the involvement of interleukin-1β (IL-1β) and tumor necrosis factor-α (TNF-α) [64]. On the other hand, 1,25(OH)2D modulates the synthesis of immunological mediators [65]. It was shown that 1,25(OH)2D and its analogue, EB1089, inhibited proliferation of the TNBC cell lines SUM-229PE and HCC1806 that expressed VDR, IL-1β, and TNF-α receptors [66]. These studies also showed that synthesis of IL-1β and TNF-α was stimulated by 1,25(OH)2D and EB1089. Moreover, 1,25(OH)2D combined with TNF-α induced a greater antiproliferative effect than either of these compounds separately. In summary, this study showed that 1,25(OH)2D induced antiproliferative effects that were underlined by the synthesis of IL-1β and TNF-α in TNBC cells. Therefore, vitamin D may exert immunomodulatory and antiproliferative effects in TNBC tumors.

Mammary stem cells are considered progenitors to breast cancer stem cells (BCSCs) responsible for breast cancer maintenance and therapy resistance [67]. Wahler et al., used the MCF10DCIS mammosphere cell culture system, enriched for mammary progenitor cells and potential BCSCs [68]. They showed that 1,25(OH)2D and BXL0124, a Gemini analog of vitamin D, decreased the efficacy of mammosphere formation, and repressed markers of stem cell-like phenotypes, including D44, CD49f, c-Notch1, and pNFκB. Moreover, these compounds decreased the expression of OCT4 and KLF-4, pluripotency markers, in mammospheres. This work showed the potential of vitamin D to prevent breast cancer through alterations in mammary stem cells. The same laboratory showed later that 1,25(OH)2D and BXL0124 decreased the level of pluripotency markers OCT4, CD44 and LAMA5 in mammospheres of the TNBC cancer cell line, SUM159 [69]. 1,25(OH)2D and BXL0124 downregulated Notch1-3, JAG1-2, HES1 and NFκB, which are important for the maintenance of BCSCs. These results suggest that some vitamin D compounds may have the potential to prevent triple-negative breast cancer through the regulation of BCSCs. Zheng et al., showed that 1,25(OH)2D increased the sensitivity of the stem cell subpopulation obtained from the MCF-7 cell line to tamoxifen [70].

### 4.3. Metabolism

It was observed that 1,25(OH)2D induced the expression and increased activity of glucose-6-phosphate dehydrogenase (G6PD), an enzyme of the pentose phosphate pathway in the TNBC MDA-MB-231, MDA-MB-468, and HCC-1143 cell lines [71]. 1,25(OH)2D reduced respiration in all these TNBC cell lines and increased citrate levels and ATP production in MDA-MB-231 cells. These and other changes induced by 1,25(OH)2D were classified as pro-survival and they did not decrease the efficacy of anticancer drugs, but only in the non-TNBC MCF-7 breast cancer cell line. The 1,25(OH)2D treatment of MDA-MB-231 cells resulted in the activation of cyclic AMP (cAMP)-dependent protein kinase (AMPK) signaling and had no effect on thioredoxin-interacting protein (TXNIP), which is a vitamin D-upregulated protein-controlling redox balance and glucose uptake protein [72,73]. However, 1,25(OH)2D reduced TXNIP in MCF-7 cells by its proteasomal degradation, which is likely endoplasmic reticulum (ER)-dependent. In summary, 1,25(OH)2D may affect TNBC cells through its various metabolic pathways in a TNBC-specific way, which is likely determined by the lack of hormone receptors.

Richards et al., studied the effect of 1,25(OH)2D and 25(OH)D on the growth of MCF-7 and three TNBC cell lines [74]. They observed that high concentrations of either compound inhibited MCF-7 growth, but did not influence the growth of TNBC cells. They also observed a substantial increase in the expression of CYP24A1 mRNA, but no systematic changes in CYP27B1 or VDR mRNA expressions. These studies present rather weak evidence that TNBC cells may be resistant to growth inhibition by vitamin D, resulting from its inactivation by CYP24A1. It may be speculated that specific metabolic pathways of the cytochrome P450 may be involved in the antiproliferative effects of vitamin D in breast cancer cells.

### 4.4. Epithelial–Mesenchymal Transition

Epithelial–mesenchymal transition (EMT) is a critical event in the invasion and metastasis of epithelium-derived carcinomas, including TNBC [75]. In general, EMT in breast cancer is associated with acquiring a stem cell-like phenotype by cancer cells [76]. Vitamin D may regulate EMT and, conversely, several EMT inducers may influence vitamin D signaling. There are many pathways of this reciprocal regulation, but E-cadherin seems to be especially important, as its downregulation is a hallmark of EMT (reviewed in [77]). There are some contradictory results on the utility of E-cadherin as a molecular marker in breast cancer as its reduced level is a potential marker of metastasis, although it is not associated with overall survival [78]. E-cadherin is essential for cell-to-cell adhesion so it is important for lymph node metastasis occurring with a higher ratio in TNBC than other types of breast cancer and it was even postulated as an independent prognostic marker in TNBC [79]. In fact, only a subset of TNBC cases show low E-cadherin expression (metaplastic carcinomas) [75,80,81]. Interleukin 6 (IL-6) may induce EMT and change the phenotype of breast cancer cells towards stemness [82]. Moreover, it was suggested that the growth of TNBC cells depended on the autocrine production of IL-6 and IL-8 [83].

It was observed that the TNBC cell line HCC1806 showed an increased expression of E-cadherin and a reduction in the subpopulation of cells with the expression of CD44, which is a marker of stemness, in incubation with 1,25(OH)2D [84]. Furthermore, transfection of TNBC cells with the *VDR* gene and incubation with 1,25(OH)2 combined with IL-6 downregulated the expression of the E-cadherin gene and increased the population of cells with CD44 expression. To explain these results the authors considered three possible pathways: demethylation of the E-cadherin promoter, inhibition of EMT-inducers and inhibition of the expression of the transcription factors important for EMT [77]. The presence of the pro-inflammatory cytokine IL-6 suppressed the anticancer activity of 1,25(OH)2D, which is in line with an earlier suggestion to this effect [85]. The main conclusion from this research is that 1,25(OH)2D may inhibit EMT and reduce stemness in TNBC cells.

Recently, Ricca et al., showed that transforming growth factor beta (TGF-β) induced the expression of VDR and that 1,25(OH)2D contrasted the TGF-β-driven EMT by transcriptional modulation [86]. The inhibitory effect of 1,25(OH)2D on EMT transition was observed only when 1,25(OH)2D was given before or along with TGF-β, but not after it. These studies were performed in human bronchial cells and cannot be directly extrapolated on TNBC cells. The inhibitory effect of 1,25(OH)2D on EMT stimulated by TGF-β in human bronchial epithelial cells was observed earlier by Fischer and Agrawal [87].

### 4.5. Epigenetics

Lopes et al., observed that 1,25(OH)2D promoted differentiation in the TNBC MDA-MB-231 cells [88]. This effect was associated with the increase in the expression of E-cadherin, an epithelial differentiation marker, and was dependent on the presence of VDR. The transcription of the cadherin 1 (*CDH1*) gene in MDB-MB-231 cells is silenced by DNA methylation in the promoter of this gene. The authors used two modifiers of the epigenetic profile: 5-aza-2′-deoxycytidine (5-aza-dC), which is a DNA demethylating agent, and trichostatin A (TSA), a histone deacetylase inhibitor (HDACi). They noted that, while the expression of the *CDH1* gene increased two to three times upon treatment with either modifier, the combination of 1,25(OH)2D with either 5-aza-dC or TSA resulted in an additive effect of *CDH1* expression both on mRNA and protein levels. Importantly, the authors observed that 1,25(OH)2D induced not only the increased expression of E-cadherin, but also its proper membrane localization, which is essential for its properties as a cell–cell adhesion molecule. Altogether, these results show that 1,25(OH)2D can act as a differentiating and DNA-demethylating agent in TNBC cells and may promote the expression of the adhesion molecule E-cadherin, and thus can be considered as a therapeutic agent in TNBC with a potential antimetastatic effect. Therefore, 1,25(OH)2D may decrease the metastatic potential of TNBC cells through the changes in their epigenetic profile.

1,25(OH)2D regulated the expression of *p21* in the breast cancer MDA-MB453 cell line through a mechanism including histone acetylation and methylation resulting in cyclical chromatin looping and approaching three VDREs in the *p21* gene to its transcription start site [89]. This led to the overproduction of p21 mRNA, which may result in a better control over cancer cells by p53.

Several types of TNBC cells are resistant to many compounds, including 1,25(OH)2D. Bijan et al., showed that such resistance in the 4T1 mouse model of TNBC might be broken by hybrid secosteroidal and non-secosteroidal analogues of 1,25(OH)2D [90]. Moreover, these analogues combined the agonism for VDR and the histone deacetylase inhibitors, as the 4T1 cells were also resistant to HDACi suberoylanilide hydroxamic acid (SAHA), but the combination of 1,25(OH)2D with SAHA induced cytostatic and cytotoxic effects in these cells. However, new analogues of 1,25(OH)2D proved to be more effective than their parent compound in reducing tumor burden and lung metastasis in 4T1 mice. These results show the therapeutic potential of secosteroidal and non-secosteroidal hybrids in TNBC and suggest further works on more effective analogues of 1,25(OH)2D.

Micro RNAs (miRNAs) are an important element of the regulation of gene expression. It was shown that the overexpression of miR-214 decreased VDR-mediated signaling in breast cancer cell lines [91]. Moreover, a positive correlation between VDR level and an inhibitor of the hedgehog pathway in breast cancer was observed. Mohri et al., showed that miR-125b regulated the expression of human VDR through interaction with the miR-125b recognition element present in the 3′-untranslated region of the *VDR* gene [92]. These authors demonstrated that the overexpression of miR-125b decreased the level of VDR in breast cancer MCF-7 cells. Moreover, 1,25(OH)2D induced cytochrome P450 family 24 subfamily A (CYP24A), but this induction was hampered by an excess of miR-145b in MCF-7 cells. Therefore, miR-145b may be targeted to potentiate the anticancer effects of 1,25(OH)2D in breast cancer. In line with these effects are recent results showing that 1,25(OH)2D inhibits the expression of miR-125b in MCF-7 cells [93]. MicroRNAs are useful as markers of both histopathological type and prognosis in TNBC [94,95,96]. Therefore, miRNAs may be important in treatment strategies for TNBC. Not only miRNAs, but also long non-coding RNAs (lncRNAs) may be important in TNBC pathogenesis and can be targeted in TNBC therapy [97,98]. Some of these lncRNAs exert their effect in breast cancer through their interaction with microRNAs [99,100]. Some lncRNAs may regulate vitamin D metabolism and 1,25(OH)2D controls the expression of some miRNAs, which, in turn, regulate the mRNAs of genes involved in breast carcinogenesis [101,102,103].

### 4.6. Therapeutic Resistance

Triple-negative breast cancer represents a therapeutic challenge due to the absence of ER, PR, and HER2 expression and, in consequence a lack of defined molecular targets and high insensitivity to tamoxifen, aromatase inhibitors and HER2-targeted therapies [104]. Accordingly, anthracyclines and taxanes are a routine therapy for TNBC, but multidrug resistance (MDR) is frequently developed in TNBC patients [105,106]. This kind of drug resistance in breast cancer is associated with several biochemical changes, including yjr overexpression of numerous proteins, such as ATP-binding cassette transporters, P-glycoprotein (P-gp), multidrug resistance-associated proteins MRP1 and MRP2 and breast cancer resistance protein (BCRP) [107]. Vitamin D was shown to modulate drug resistance in cancer in numerous pathways (reviewed in [108]). Specifically, Guo et al., showed that vitamin D in combination with metformin inhibited the proliferation of human breast cancer MDA-MB-231 cells and promoted apoptosis [109]. It was also shown that vitamin D improved the chemotherapeutic efficacy of cyclooxygenase-2 (COX-2) inhibitors in MDA-MB-231 breast cancer cells [110,111].

One of the successful strategies to overcome cancer drug resistance is the application of nanoscale drug delivery carriers [112]. Kutlehria et al., designed a conjugate of cholecalciferol glutarate with poly (ethylene glycol) 2000 (PEGCCF) [113]. This conjugate self-assembled to form micelles that could be loaded with doxorubicin (DOX) and improved the chemotherapeutic potential of this anthracycline. It was shown that PEGCCF increased DOX accumulation in MDA-MB-231 cells through the inhibition of P-gp. The use of the micelles enhanced the cytotoxicity of DOX in TNBC DOX-resistant cell lines (MDA-MB-231DR) almost three times. It was shown that PEGCCF–DOX caused the significant downregulation of mechanistic target of rapamycin (mTOR), c-master regulator of cell cycle entry and proliferative metabolism (c-Myc), and antiapoptotic B-cell lymphoma-extra large (Bcl-xL) along with the upregulation of preapoptotic Bcl2 X associated (Bax) as well as ATP binding cassette subfamily B member 1 (ABCB1) downregulation, enhanced chemosensitization and apoptosis.

Santagata et al., observed that about two thirds of TNBC patients expressed VDR and/or androgen receptor (AR) [114]. In their subsequent work, Thakkar et al., identified two TNBC cell lines with the expression of both receptors and checked their phenotype, viability and proliferation after treatment with VDR or AR agonist 1,25(OH)2D or dihydrotestosterone, respectively [115]. Both kinds of agonist hormones decreased the viability of TNBC cells and their joint action was additive. The effect was potentiated when agonists were combined with chemotherapy. The inhibition of viability was associated with inhibition in the cell cycle and apoptosis in TNBC cells. In addition, these agonist hormones induced the differentiation of cancer stem cells. Therefore, agonist hormone therapy targeting VDR and AR may be effective in TNBC and may be considered to be applied in combination with conventional chemotherapy.

Paclitaxel (PTX), a taxane, belongs to the most widely used anticancer drugs and is often applied in front-line chemotherapy in breast cancer in combination with anthracyclines or platinum chemotherapeutic drugs (reviewed in [116]). Wilhelm et al., showed that 1,25(OH)2D and 25(OH)D at low (up to 10 nM) concentrations increased the proliferation of the p53-wild type TNBC DU4475 cells and potentiated the cytotoxic action of PTX [117]. However, this effect was not observed in MDA-MB-231 cells that did not contain p53. This difference was not clearly explained in that study because, as well as the difference in p53 expression, these cell lines belong to different subclasses of TNBC and this may determine their responsiveness in many ways. Other reasons for that difference are also possible and should be addressed in further research.

## 5. Critical Role of BRCA1 and TP53BP1 in VD3 Signaling in Triple-Negative Breast Cancer

About 5% of all breast cancer cases are associated with pathogenic variants of DNA repair-associated breast cancer type 1 (*BRCA1*) susceptibility, and *BRCA2* genes [118]. These variants increase the lifetime risk of breast cancer by 40%–90% [119]. Products of both these genes are involved in DNA repair, mainly in homologous recombination repair [120]. Many breast cancer cases associated with *BRCA1* mutations are classified as TNBC as they do not show the expression of ER, PR and HER2. This kind of breast cancer is usually highly aggressive and difficult to treat [121]. A subset of sporadic TNBCs have defects in DNA repair and gene expression profiles typical of BRCA1-related cancers and they can be treated with therapeutic strategies based on deficiencies in DNA repair [122]. BRCA1-deficient TNBCs are prone to poly(ADP-ribose) polymerase inhibitors (PARP_i_), whose action results in the deficient repair of DNA single-strand breaks (SSBs), which, when unrepaired, produce DSBs in replicating cancer cells [123,124]. Therefore, the loss of BRCA1 is synthetically viable with the loss of PARP and dictates a therapeutic strategy in BRCA1-deficient TNBCs [125].

Campbell et al., showed that 1,25(OH)2D induced the expression of BRCA1 mRNA in the vitamin D-sensitive breast cancer cell line MCF-7, but not in their vitamin D-resistant counterpart MDA-MB-436 [126]. On screening several vitamin D-sensitive and -resistant breast cancer cell lines, these authors suggested that the sensitivity to the antiproliferative action of 1,25(OH)2D was strongly associated with its ability to modulate BRCA1. Moreover, vitamin D sensitivity correlated with ER expression. Therefore, VDR may induce factors that transactivate the *BRCA1* gene and its expression mediates the antiproliferative effect of 1,25(OH)2D, but this pathway may be disrupted in breast cancer by pathogenically mutated *BRCA1* or/and aberrant VDR signaling. In turn, Graziano et al., showed that RAS-induced senescence in human immortalized cell lines was associated with the downregulation of VDR and the vitamin D/VDR axis regulated the expression of the *BRCA1* gene [127].

Cysteine proteases are proteolytic enzymes that promote cancer invasion and metastasis as they degrade basement membrane and extracellular matrix. Cathepsin L (CTSL) is likely the most widely studied enzyme of this group and it is targeted in anti-metastatic therapy [128]. It is also important for the angiogenesis required by both primary and metastatic tumors [129]. Moreover, CTSL can contribute to surface characteristics of cancer cells, which can be important in distinguishing cancer cells from their normal counterparts [130]. However, it was also shown that CTSL might be involved in the regulation of cell cycle progression through proteolytic processing of the cut-like homeobox 1 (CUX-1) transcription factor [131,132].

Burton et al., showed that CTSL, along with its downstream substrate CUX1, were overexpressed in samples from TNBC patients and TNBC cell lines in comparison to their ER-positive counterparts [133]. The inhibition of CUX1 in various TNBC cell lines by a CTSL inhibitor decreased the migration and invasiveness of these cells. Muscadine grape skin extract (MSKE) inhibited CUX1 expression, decreased its binding to the promoter of the *ER-α* gene and restored the expression of *ER-α* in the TNBC MDA-MB-468 cells. Both MSKE and CUX1 siRNAs restored the sensitivity of these cells to estradiol and 4-hydroxytamoxifen. Therefore, CTSL signaling can be involved in TNBC pathogenesis through CUX1 activation and CTSL and CUX1 may be targeted in TNBC therapy.

Germline mutations in the *BRCA1* gene occurring in breast cancer cases often result in a complete loss of function of BRCA1 [134,135]. In turn, BRCA1 loss is synthetically viable with the loss of tumor protein p53-binding protein 1 (TP53BP1) [136]. Both proteins are decisive in DNA double-strand break repair (DSBR) choice—BRCA1 promotes homologous recombination repair (HRR), whereas TP53BP1 promotes non-homologous end joining (NHEJ), although these relationships may change over time [137]. In general, the loss of TP53BP1 is associated with a worse prognosis in breast cancer and consequently TP53BP1 is considered as a tumor suppressor [138]. Gonzalo’s lab demonstrated that 1,25(OH)2D stabilized TP53BP1 via its interaction with VDR and promoted DSBR through the inhibition of CTSL, which is important in cancer progression and is involved in TP53BP1 degradation [139]. This effect might be mediated by endogenous inhibitors of cathepsins, as it was shown that 1,25(OH)2D upregulated cystatin D, which inhibited cathepsin D in the colorectal cancer cell line [140]. The upregulation of CTSL and its accumulation in the nucleus were found to be associated with the loss of A-type lamins [139]. However, such a loss resulted in a decrease in BRCA1 and RAD51 on both mRNA and protein levels [141]. Therefore, it could be speculated that 1,25(OH)2D and cathepsin inhibitors might rescue the levels of BRCA1 and RAD51, an assumption that was supported by the observations that 1,25(OH)2D reduced the basal level of DNA damage, the morphological defect characteristics of A-type lamins and the VDR-mediated expression of the *BRCA1* gene [126]. The experiments in Gonzalo’s lab showed that 1,25(OH)2D rescued the levels of TP53BP1 and its ability to localize at the site of damaged DNA, as well as an important role of 1,25(OH)2D in the regulation of the two main DSB repair pathways, HRR and NHEJ [142] (Figure 2).

Work from Gonzalo’s group showed that 1,25(OH)2D increased genomic instability after ionizing radiation exposure in BRCA1-deficient cells that overcame cell arrest (BOGA) [143]. These results suggest a possible therapeutic strategy in breast cancer based on the induction of radiosensitization in BRCA1-deficient cells that activate the CTSL-mediated degradation of TP53BP1. However, there are some limitations to this strategy: firstly, it could be applied to tumors that have activated CTLS-mediated degradation of TP53BP1 and, secondly, these cells should be identified, which may be challenging [142]. As most of the action of 1,25(OH)2D requires a functional nuclear VDR [144], it was hypothesized that the upregulation of nuclear VDR might result in the activation of cystatins and the inhibition of CTSL-mediated TP53BP1 degradation [142]. Due to this hypothesis, high levels of VDR might be typical for breast tumors with high level of nuclear CTLS and TP53BP1. This was supported by experimental data showing a strong positive correlation between low levels of VDR and a decrease in TP53BP1 dependent on an increase in nuclear CTSL [142]. This relationship was especially strong in TNBC tumors. The exact mechanism by which vitamin D inhibits CTSL is not completely known, but it can be mediated by cystatins, including cystatin A [145]. In summary, 1,25(OH)2D may protect against breast cancer by activating its receptor and inactivating the CTSL-mediated degradation of TP53BP1 in A-type lamin-deficient cells, which display BRCA1 deficiency, including cells with lost BRCA1. Therefore, some breast cancer cases carry a triple biomarker signature—the levels of nuclear VDR, CTSL and TP53BP1 that may be exploited in projecting a treatment strategy in TNBC cases.

Heublein et al., studied the expression of VDR in breast cancer patients with mutated (*n* = 38) and non-mutated *BRCA1* genes (*n* = 79) [146]. They observed that VDR was detected in over 90% of BRCA1-mutated TNBC cases and was overexpressed in individuals with mutated *BRCA1* in comparison with their non-mutated counterparts. Similar results were obtained for other receptors: retinoid X receptor (RXR) and peroxisome proliferator-activated receptor γ (PPARγ). These three receptors interact with thyroid hormone receptors (TRs), forming a functional heterotetramer. This study confirms the importance of BRCA1 in vitamin D signaling in TNBC.

## 6. GADD45A—a New Player in Vitamin D Signaling in Triple-Negative Breast Cancer

The growth arrest and DNA damage-inducible 45 alpha (GADD45A) gene is important in the cellular reaction to stress as it is upregulated in stress conditions induced by various factors, including DNA-damaging agents [147].

Tront et al., observed the suppressive role of GADD45A in a mouse model of breast cancer driven by Ras activation [148]. In addition, these authors showed that this Ras-driven tumor formation in the absence of GADD45A resulted in a decrease in apoptosis and senescence. These effects were mediated by c-Jun N-terminal kinase (JNK) and p53, respectively, linked to a decrease in c-Jun NH(2)-terminal kinase (JNK) activation, and a decrease in Ras-induced senescence, correlating with a decrease in p38 kinase activation. Therefore, the protective action of GADD45A in Ras-driven breast cancer may include its cytotoxic and cytostatic action in breast cancer cells and so this protein can be considered a target in breast cancer therapy.

Immunohistochemical detection of GADD45A in normal and breast tissue samples revealed that its level was strongly associated with hormone receptor status in human breast cancer [149]. Normal breast tissue displayed a low GADD45A level, whereas luminal A and luminal B subtypes were characterized by high levels of GADD45A, but TNBC tumors were negative for or had low levels of this protein. Similar studies in 419 breast cancer samples and 116 adjacent non-cancerous tissue revealed that this protein was overexpressed in patients with a worse prognosis in TNBC [150]. Therefore, GADD45A expression level may be useful in stratifying TNBC patients for treatment.

Flores et al., observed that 1,25(OH)2D inhibited the growth of epithelial cells derived from prostate tumor with concomitant upregulation of GADD45γ [151]. Upregulation of GADD45γ was independent of androgen signaling. Taken together, this study established GADD45γ as a growth-inhibitory protein in prostate cancer and indicated it as a possible therapeutic target in this disease as it may be stimulated by 1,25(OH)2D.

In summary, many studies indicate that 1,25(OH)2D may inhibit proliferation of many kinds of cancer cells through molecular pathways including G1 and G2/M arrests by the activation of all three members of the GADD45 proteins family [152,153,154,155,156,157]. Many tumors and cancer cell lines have low levels of GADD45A and other members of the GADD45 family and, in breast cancer cases, this relationship is especially marked in TNBC. In addition, BRCA1 may stimulate GADD45A-mediated nucleotide excision repair and DNA demethylation in breast cancer (reviewed in [158]). Therefore, GADD45A, and likely two other members of the GADD45 family, may be important in TNBC pathogenesis and thus could be considered in the therapy of this disease.

## 7. Conclusions and Perspectives

Vitamin D is known to display anticancer properties, but many aspects of these properties remain unknown or are problematic, if not conflicting. This also concerns the action of vitamin D in breast cancer. However, most of the controversies result from regarding breast cancer as a single disease, while it can occur in many distinct forms. Triple-negative breast cancer, which belongs to the category of the most difficult to handle cancers, can be featured by at least six distinguished molecular characteristics. Therefore, in light of studies that have been performed so far, it may be concluded that the discrepancy in the results of studies on protective potential of vitamin D in TNBC may lie in the enormous diversity of TNBC tumors and cells. Therefore, the corresponding perspective is to design studies on vitamin D and TNBC with a homogenous population of patients or TNBC cell lines with exact molecular characteristics.

Although several studies pointed out that 25(OH)D concentrations close to physiological ones may play some beneficial role in breast cancer prevention, it seems doubtful that dietary supplementation with vitamin D may be adopted as a preventive or therapeutic strategy against TNBC and by no means should be recommended; the more negative effects of vitamin D that were not addressed in this review may be quite serious (reviewed in [159]).

As 1,25(OH)2D can be involved in the regulation of miRNA expression, the investigation of the mRNA–miRNA–lncRNA regulatory axis for the genes important in breast cancer is justified.

The *VDR* gene has several polymorphisms that have been associated with the occurrence of many disorders (reviewed in [160]). However, in breast cancer there are some contradictory results or results indicating no association (reviewed in [161]). As mentioned earlier, there is a substantial difference in the ratio of breast cancer between black and white women. Amadori et al., studied vitamin D concentration in serum and the 1012A>G, Cdx2 and Fok1 polymorphisms of the VDR gene in healthy African and Caucasian women and those with breast cancer. They found that healthy African women had lower vitamin D levels than their Caucasian counterparts and they had a higher ratio of the AA and CC genotypes of the Cdx2 and Fok1 polymorphism, respectively. Therefore, the variability of the VDR gene may contribute to the ethnic differences in breast cancer occurrence in correlation with serum vitamin D content. However, the Fok1, Bsm1, Taq1, and Apa1 polymorphisms of the VDR gene were not associated with breast cancer in a white population [162].

Multi-pathway mechanisms may be involved in the protective action of vitamin D against TNBC, including the modulation of oxidative stress, antiproliferative action, the modification of epigenetic profile and the differentiation of TNBC cells. However, many details of these mechanism are yet to be determined. The immunomodulatory effect of vitamin D in breast cancer in general and in TNBC in particular was presented in the previous sections. However, it is commonly accepted that innate and adaptive immune cells infiltrating malignant tumors are associated with clinical outcomes and responses to treatment. This is an emerging issue that has initiated the immuno-oncology era, with hope for a breakthrough in cancer therapy (reviewed in [163]). This seems especially important in light of the current COVID-19 pandemic. Therefore, further studies are justified to establish the potential of vitamin D in immuno-oncology of breast cancer, especially given that the presence of VDR was found in most, if not all, immune cells (reviewed in [164]). Moreover, defects in vitamin D are observed in several autoimmunological diseases [165]. Apart from the mechanisms presented in the previous sections, vitamin D may modulate the damage-associated molecular pattern signaling involved in cancer transformation [166,167]. This can be underlined by the upregulation of IL-10 and T regulatory lymphocytes, with concomitant blocking of T helper cytokines, including IL-2, IL-32, interferon-γ, and IL-17 (reviewed in [14]).

These TNBC tumors that bear pathogenic mutation(s) in the *BRCA1* gene may be especially difficult to cure due to their aggressiveness and therapy resistance. However, thanks to the work of Gonzalo’s lab and others, we can draw some conclusions about the mechanisms of the pathogenesis of this type of cancer. These TNBC tumors are lamin-deficient, which contributes to genomic instability through a proteolytic degradation of the TP53BP1 protein by cathepsin L and the inhibition of RAD51, resulting in aberrant repair of DNA double-strand breaks by NHEJ and HRR. Vitamin D may activate its receptor, VDR, and stimulate a cystatin to inhibit cathepsin L and prevent the proteolytic degradation of TP53BP1 and, consequently, protect NHEJ and therefore genomic stability. These studies might be extended to explore another DNA double-strand repair system, single-strand annealing and its main protein, RAD52 (reviewed in [168]).

The expression of the *BRCA1* gene or a lack thereof may be important in breast cancer pathogenesis, as outlined in Section 5. When the *BRCA1* gene is normally expressed, resulting in the normal BRCA1 protein, its stability is critical as it ensures that a normal level of the gene is present, which is necessary for its normal functions, thus preventing breast carcinogenesis. We cannot find evidence on studies about the influence of vitamin D on the stability of BRCA1, but (apparently) this problem should be addressed in further research.

The GADD45 family of proteins may play an important role in breast carcinogenesis, especially due to the fact that BRCA1 may stimulate GADD45A-mediated DNA repair and GADD45A may regulate vitamin D signaling.

Vitamin D may have the potential to exert a beneficial effect on the pathogenesis and therapy of TNBC, but the determination of the circumstances of this effect strongly depend on the molecular characteristics of TNBC cells that should be determined in further research.

## Figures and Tables

**Figure 1 ijms-21-03670-f001:**
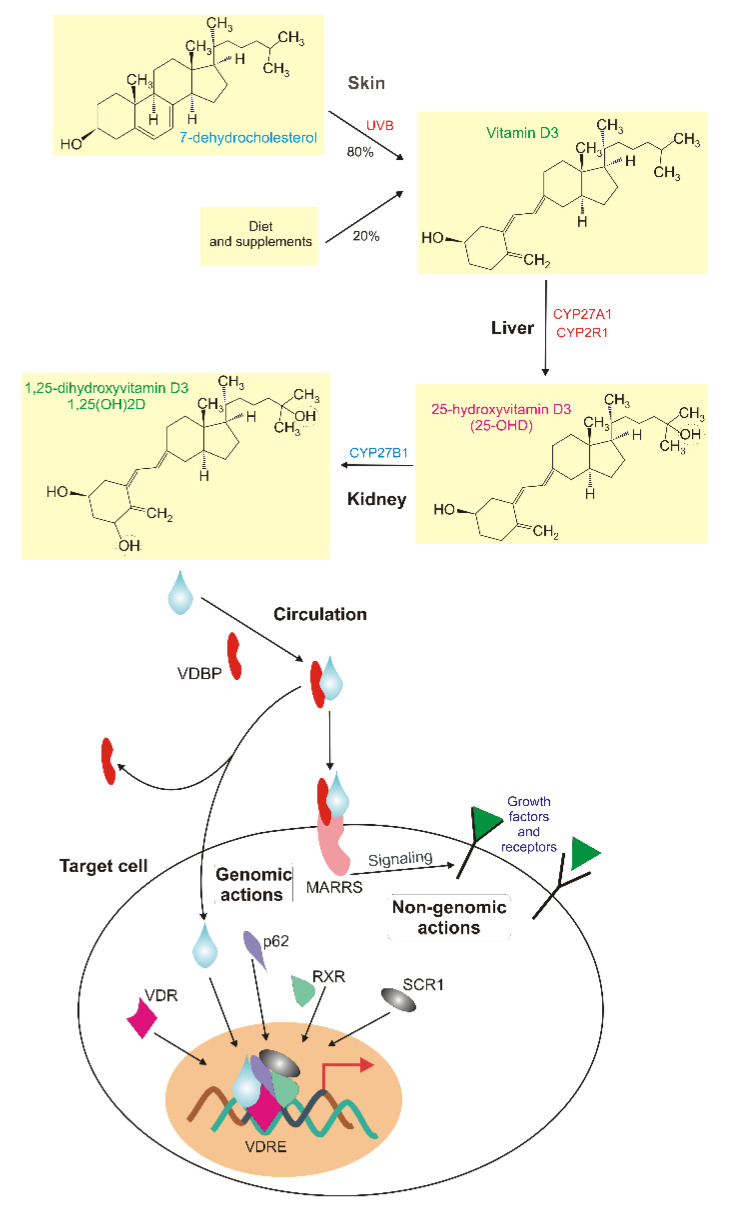
Vitamin D metabolism and effects in humans. Vitamin D3 is synthesized in skin from 7-dehydrocholesterol in a reaction catalyzed by solar UVB and can be delivered with food. In the circulation, it is bound by vitamin D binding protein (VDBP) and can reach liver, where is metabolized by cytochrome P450 family 2 subfamily R member 1 and family 27 subfamily A member 1 (CYP2R1 and CYP27A1) to 25-hydroxyvitamin D3 (25(OH)D), which is converted by CYP24A1 to 1,25-dihydroxyvitamin D3 (1,25(OH)2D, a blue tear) in a reaction that occurs mainly in kidney microtubule. 1,25(OH)2D is a biologically active metabolite of vitamin D that may interact with its transmembrane membrane-associated rapid response steroid-binding (MARRS) receptor and affect signaling, nuclear proteins and other transmembrane receptors, including growth factors (non-genomic action). 1,25(OH)2D may associate with its nuclear receptor Vitamin D receptor (VDR), which induces its heterodimerization with retinoid X receptor (RXR) and binding other proteins, including sequestosome 1 (p62/SQSTM1) and steroid receptor coactivator 1 (SRC1), to bind vitamin D response elements in the promoters of hundreds of genes to regulate their expression (genomic action).

**Figure 2 ijms-21-03670-f002:**
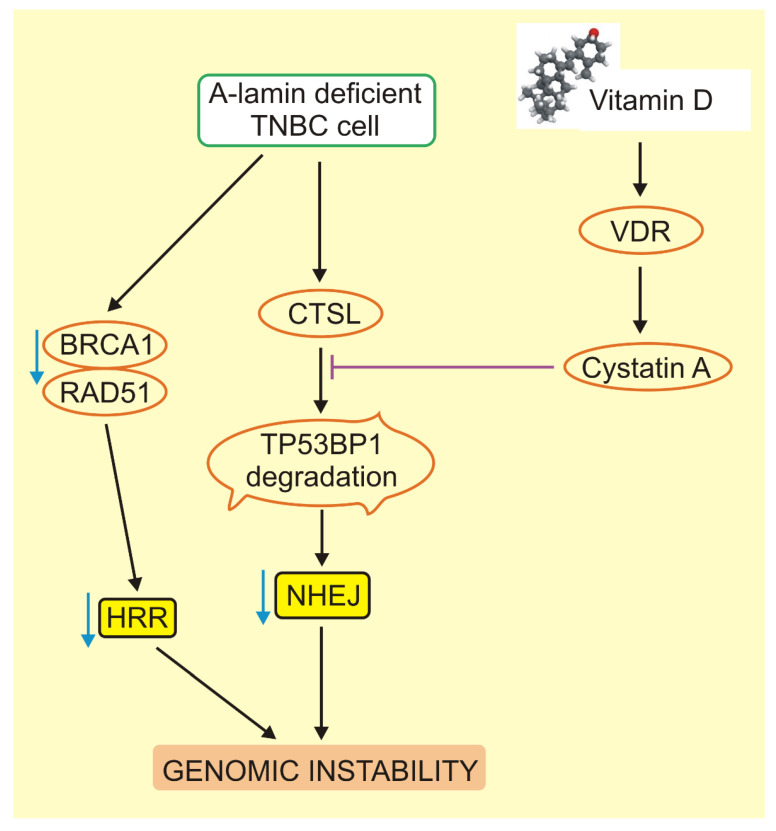
Vitamin D may ameliorate defects in DNA repair associated with the loss of A-type lamins and increased activity of cathepsin (CTSL) L in triple-negative breast cancer (TNBC). A-type and other lamins play an important role in maintaining of cellular structure and genomic stability and their deficiency may occur in TNBC resulting in a series of effects, including CTSL activation, which leads to degradation of the TP53BP1 (tumor protein P53 binding protein 1) protein that is important in non-homologous end joining (NHEJ), the main DNA double-strand break repair (DSBR) system in humans. On the other hand, the loss of A-type lamins decreases the levels of DNA repair-associated breast cancer type 1 susceptibility (BRCA1) and RAD51 (RAD51 recombinase), major proteins of the other main DSBR system in humans—homologous recombination repair (HRR). Altogether, NHEJ and HRR deficiency contributes to genomic instability. Vitamin D may activate its receptor (VDR) and stimulate transcription of cystatin D that inhibits CTSL and in this way prevents TP53BP1 degradation, NHEJ deficiency and ameliorates genomic stability. Vitamin D may also prevent BRCA1 and RAD51 repression, but the exact mechanism underlying this effect is not completely known.

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
