# Peer review of "Vitamin D in Triple-Negative and BRCA1-Deficient Breast Cancer—Implications for Pathogenesis and Therapy"

_ijms, 2020, doi:10.3390/ijms21103670_

Round 1

Reviewer 1 Report

Manuscript describing the protective role of Vitamin D in breast cancer, particularly its suppressive role in TNBC subtype, seems to bring an important discussion point to readers. However, in this era of immuno-oncology (IO) and even further during this devastating COVID-19 pandemic, the role of vitamin D and its hormonal signaling pathways that are known to modulate both innate and adaptive immunity could have been emphasized in this particular review manuscript. It is too bad I don't see such discussion sessions in the manuscript. 

Overall, manuscript is well-structured and nicely written. However, language editing is strongly advised.  

Author Response

Comment: Manuscript describing the protective role of Vitamin D in breast cancer, particularly its suppressive role in TNBC subtype, seems to bring an important discussion point to readers. However, in this era of immuno-oncology (IO) and even further during this devastating COVID-19 pandemic, the role of vitamin D and its hormonal signaling pathways that are known to modulate both innate and adaptive immunity could have been emphasized in this particular review manuscript. It is too bad I don't see such discussion sessions in the manuscript.

Answer: We have added the following fragment to 7. Conclusions and Perspectives:

“The immunomodulatory effect of vitamin D in breast cancer in general and in TNBC in particular was presented in the previous sections. However, it is commonly accepted that innate and adaptive immune cells infiltrating malignant tumors are associated with clinical outcomes and responses to treatment. This is an emerging issue that initiated the immuno-oncology era with hope for a breakthrough in cancer therapy (reviewed in [163]). This seems especially important in the current COVID-19 pandemic time. Therefore, further studies are justified to establish the potential of vitamin D in immuno-oncology of breast cancer, especially that the presence of VDR was found in most, if not all, immune cells (reviewed in [164]). Moreover, defects of vitamin D are observed in several autoimmunological diseases [165]. Apart from the mechanisms presented in the previous sections, vitamin D may modulate the damage-associated molecular pattern signaling involved in cancer transformation [166,167]. This can be underlined by upregulation of IL-10 and T regulatory lymphocytes with concomitant blocking T helper cytokines, including IL-2, IL-32, interferon-γ, and IL-17 (reviewed in [168]).”

With new references:

  1. Allard, B.; Aspeslagh, S.; Garaud, S.; Dupont, F.A.; Solinas, C.; Kok, M.; Routy, B.; Sotiriou, C.; Stagg, J.; Buisseret, L. Immuno-oncology-101: overview of major concepts and translational perspectives. Seminars in cancer biology 2018, 52, 1-11, doi:10.1016/j.semcancer.2018.02.005.
  2. Chirumbolo, S.; Bjørklund, G.; Sboarina, A.; Vella, A. The Role of Vitamin D in the Immune System as a Pro-survival Molecule. Clinical therapeutics 2017, 39, 894-916, doi:10.1016/j.clinthera.2017.03.021.
  3. Altieri, B.; Muscogiuri, G.; Barrea, L.; Mathieu, C.; Vallone, C.V.; Mascitelli, L.; Bizzaro, G.; Altieri, V.M.; Tirabassi, G.; Balercia, G., et al. Does vitamin D play a role in autoimmune endocrine disorders? A proof of concept. Reviews in Endocrine and Metabolic Disorders 2017, 18, 335-346, doi:10.1007/s11154-016-9405-9.
  4. Eppensteiner, J.; Davis, R.P.; Barbas, A.S.; Kwun, J.; Lee, J. Immunothrombotic Activity of Damage-Associated Molecular Patterns and Extracellular Vesicles in Secondary Organ Failure Induced by Trauma and Sterile Insults. Front Immunol 2018, 9, doi:10.3389/fimmu.2018.00190.
  5. Hernandez, C.; Huebener, P.; Schwabe, R.F. Damage-associated molecular patterns in cancer: a double-edged sword. Oncogene 2016, 35, 5931-5941, doi:10.1038/onc.2016.104.
  6. Pandolfi, F.; Franza, L.; Mandolini, C.; Conti, P. Immune Modulation by Vitamin D: Special Emphasis on Its Role in Prevention and Treatment of Cancer. Clinical therapeutics 2017, 39, 884-893, doi:10.1016/j.clinthera.2017.03.012.

Comment: Overall, manuscript is well-structured and nicely written. However, language editing is strongly advised.

Answer: We have done our best to improve the language quality of the manuscript.

Reviewer 2 Report

The authors have provided quite extensive review on vit D and cancer with focus on TNBC. The referrence list is very good as well.

  1. Can they provide a table for clinical work on this with the references that point to relevant citations? just to separate from mechanistic study on cell lines.
  2. what do they think about VDR polymorphisms? since significant amount of work has  been done with respect to cancer risk.
  3. line 240-241, does this mean low vit. d level good for prognosis? can they clarify?
  4. Is there any relation beween BRCA1 protein stability and VDR/VitD? it can be discussed?

Author Response

The authors have provided quite extensive review on vit D and cancer with focus on TNBC. The referrence list is very good as well.

Comment: Can they provide a table for clinical work on this with the references that point to relevant citations? just to separate from mechanistic study on cell lines.

Answer: The primary goal of this manuscript was to present molecular aspects of vitamin D in breast cancer with a special emphasis of its triple-negative form. However, our manuscript contains some data on epidemiological works and clinical trials – they are presented in section 3. Vitamin D in Breast Cancer and subsection 4.2 Observational Studies. They are arbitrarily chosen on the basis or their relation to the main subject of the manuscript. Some of them present results of meta-analyses, containing many observational studies. Including only these that are directly or indirectly mentioned in the manuscript would extend it dramatically as a huge table with about 100 references would be added. Therefore, we suggest not to follow this idea – it would be beneficial in a more clinically-oriented manuscript.

Comment: what do they think about VDR polymorphisms? since significant amount of work has  been done with respect to cancer risk.

Answer: We have added the following fragment to 7. Conclusions and perspectives section:

“The VDR gene has several polymorphisms that have been associated with the occurrence of many disorders (reviewed in [160]). However, in breast cancer there are some contradictory results or results indicating no association (reviewed in [161]). As mentioned earlier, there is a substantial difference in the ratio of breast cancer between black and white women. Amadori et al. studied vitamin D concentration in serum and the 1012A>G, Cdx2 and Fok1 polymorphisms of the VDR gene in African and Caucasian healthy and breast cancer women. They found that healthy African women had lower vitamin D levels than their Caucasian counterparts and they had a higher ratio of the AA and CC genotypes of the Cdx2 and Fok1 polymorphism, respectively. Therefore, the variability of the VDR gene may contribute to the ethnic differences in the breast cancer occurrence in correlation with serum vitamin D content. However, the Fok1, Bsm1, Taq1, and Apa1 polymorphisms of the VDR gene were not associated with breast cancer in a white population [162]”

with new references:

  1. Valdivielso, J.M.; Fernandez, E. Vitamin D receptor polymorphisms and diseases. Clinica chimica acta; international journal of clinical chemistry 2006, 371, 1-12, doi:10.1016/j.cca.2006.02.016.
  2. Li, J.; Li, B.; Jiang, Q.; Zhang, Y.; Liu, A.; Wang, H.; Zhang, J.; Qin, Q.; Hong, Z.; Li, B.A. Do genetic polymorphisms of the vitamin D receptor contribute to breast/ovarian cancer? A systematic review and network meta-analysis. Gene 2018, 677, 211-227, doi:10.1016/j.gene.2018.07.070.
  3. Lu, D.; Jing, L.; Zhang, S. Vitamin D Receptor Polymorphism and Breast Cancer Risk: A Meta-Analysis. Medicine 2016, 95, e3535, doi:10.1097/md.0000000000003535.

Comment: line 240-241, does this mean low vit. d level good for prognosis? can they clarify?

Answer: No, it is written that high level of vitamin D is associated with lower breast cancer risk, especially in patients with poor prognosis. To make this fragment more straight, we have changed the sentence:

“In several other studies higher serum levels of 25(OH)D was associated with reduced risk of breast cancer with the strongest association for types with poor prognosis, especially TNBC [49].”

Into

“Higher serum levels of 25(OH)D was associated with a reduced risk of breast cancer, especially in cases with poor prognosis, including TNBC [49].”

Comment: Is there any relation beween BRCA1 protein stability and VDR/VitD? it can be discussed?

Answer: We have added the following fragment to 7. Conclusion and Perspectives:

“Not only the expression of the BRCA1 gene or it lack may be important in breast cancer pathogenesis as outlined in section 5. Critical Role of BRCA1 and TP53BP1 in VD3 Signaling in Triple-Negative Breast. When the BRCA1 gene is normally expressed, resulting in the normal BRCA1 protein, its stability is critical as it ensures its normal level necessary to its normal functions, preventing breast carcinogenesis. We cannot find evidence on studies on the influence of vitamin D on the stability of BRCA1, but apparently this problem should be addressed in further research.”